# ROBOGait: A Mobile Robotic Platform for Human Gait Analysis in Clinical Environments

**DOI:** 10.3390/s21206786

**Published:** 2021-10-13

**Authors:** Diego Guffanti, Alberto Brunete, Miguel Hernando, Javier Rueda, Enrique Navarro

**Affiliations:** 1Centre for Automation and Robotics (CAR UPM-CSIC), Universidad Politécnica de Madrid, 28012 Madrid, Spain; alberto.brunete@upm.es (A.B.); miguel.hernando@upm.es (M.H.); 2Universidad Tecnológica Equinoccial (UTE), 230208 Santo Domingo, Ecuador; 3Department of Human Health and Performance, Faculty of Sports Sciences, Universidad Politécnica de Madrid, 28040 Madrid, Spain; javier.ruedao@alumnos.upm.es (J.R.); enrique.navarro@upm.es (E.N.)

**Keywords:** human gait analysis, mobile robotic platforms, clinical environments, multiple sclerosis, markerless system, motion capture

## Abstract

Mobile robotic platforms have made inroads in the rehabilitation area as gait assistance devices. They have rarely been used for human gait monitoring and analysis. The integration of mobile robots in this field offers the potential to develop multiple medical applications and achieve new discoveries. This study proposes the use of a mobile robotic platform based on depth cameras to perform the analysis of human gait in practical scenarios. The aim is to prove the validity of this robot and its applicability in clinical settings. The mechanical and software design of the system is presented, as well as the design of the controllers of the lane-keeping, person-following, and servoing systems. The accuracy of the system for the evaluation of joint kinematics and the main gait descriptors was validated by comparison with a Vicon-certified system. Some tests were performed in practical scenarios, where the effectiveness of the lane-keeping algorithm was evaluated. Clinical tests with patients with multiple sclerosis gave an initial impression of the applicability of the instrument in patients with abnormal walking patterns. The results demonstrate that the system can perform gait analysis with high accuracy. In the curved sections of the paths, the knee joint is affected by occlusion and the deviation of the person in the camera reference system. This issue was greatly improved by adjusting the servoing system and the following distance. The control strategy of this robot was specifically designed for the analysis of human gait from the frontal part of the participant, which allows one to capture the gait properly and represents one of the major contributions of this study in clinical practice.

## 1. Introduction

The human gait contains important information about the health status of human beings. It can be used to monitor health problems, as in the case of neurological diseases [1]. Some studies have shown that neurological diseases, such as stroke [2], Parkinson’s disease [3], cerebral palsy (CP) [4], multiple sclerosis (MS) [5], and others, alter gait patterns. Alterations in gait patterns lead to loss of freedom of movement, increase the risk of falls and injuries, and cause a significant reduction in quality of life [6].

Motion analysis systems help to understand and quantify the evolution of these diseases by analyzing the human gait. A detailed review of the state of the art revealed the popularity as well as the accuracy of high-end optoelectronic systems, such as Vicon, Qualisys, and OptiTrack, among others. The popularity of the use of inertial systems, such as Xsens, has also been noted. It is not new that these systems require a large investment. Only hospitals with extensive financial resources can have one of these systems, making the acquisition almost impossible in clinics or rehabilitation centers.

Even though these are the gold-standard systems in gait analysis, they require the installation of a dedicated laboratory space, which is also not available in all cases. In addition, these systems use wearable methods of gait analysis, as they require the placement of markers or sensors on each joint to be analyzed. Furthermore, the use of these systems show several limitations, including long preparation times, soft tissue artifacts, or unfeasibility of specific movements due to the presence of the markers, which can hinder the correct execution of the movement [7]. Due to the long procedure time on these systems, patients are usually already fatigued once the test begins. These drawbacks can limit the use of these systems within certain areas of motion analysis.

On the contrary, fully automated and markerless systems are overcoming these drawbacks for conducting biomechanical studies, especially outside laboratories [8]. The popularity of the use of RGBD cameras for gait analysis has increased, as they are faster to implement, markerless, and cost-effective, benefiting both evaluators and patients. Several options of RGB-depth (RGBD) cameras and libraries can perform skeletal tracking applications. Among the most popular on the market are Intel Realsense (D415, D435, EUCLID), Orbbec (Persee, Astra, Astra Pro), TVICO, Kinect (V1, V2), and the most recent Azure Kinect. The evaluation of these sensors for human pose estimation [9] and human gait analysis [10] has been extensively studied by the research community.

However, one of the major limitations of RGBD cameras is their short range of view (approximately 3 m), which is not enough to analyze human gait [11]. To solve this problem, several studies have proposed the use of multiple sensors [12,13] or the use of treadmills [14,15]. Multiple sensors have proven to be a valid alternative; however, they require a dedicated laboratory and complicated calibration processes [11]. On the other hand, the use of treadmills has been shown to disturb the normal gait pattern of patients, as is stated in the study presented by Shi et al. [16].

Taking these considerations into account, the inclusion of depth sensors in clinical practice requires a more portable and more natural gait assessment solution that enables examinations, for example, in the corridors of a clinic or hospital. To solve these requirements, in this study, we propose the use of a mobile robotic platform based on depth cameras to perform the analysis of human gait in practical scenarios. The platform is able to follow the patient while evaluating the walking pattern. The measurements are more natural because these tests are performed in real environments, in contrast to the use of a dedicated laboratory.

In this context, the aim of this study is to prove the validity of this robot and its applicability in clinical settings. The system was validated in a controlled environment by comparing it with a certified system. Clinical tests with patients with MS gave an initial impression of the applicability of the instrument in patients with abnormal walking patterns. Overall, this study describes the entire path from the idea to an actually implementable instrument, whereby conditions regarding the clinical applicability and accuracy are constantly tested.

## 2. State of the Art

With the advancement of mobile robotics through the incorporation of new sensors and algorithms that are useful for navigation, it is possible to provide depth sensors with some autonomy by integrating them into robotic systems designed for gait analysis. The use of mobile robots for human gait monitoring and analysis is a field generating a great deal of interest in the medical community. Basically, there are two types of mobile robotic systems for human gait analysis. The difference between them lies in whether the patient must walk with physical human–robot interaction (pHRI) by using a parallel handle configuration mounted on the robot, or whether the patient walks at a distance away from the robot and without contact, which is known as cognitive human–robot interaction (cHRI). Platforms that require physical contact are commonly referred to as “walker robots” and have a high utility in the physical rehabilitation of patients. On the other hand, platforms that do not require physical contact combine the use of navigation and person-following algorithms. This is the reason for why these robots are also known as “follower robots”. There are algorithms in which the robot follows the person along a free walking path and others in which the person follows the robot along a predefined walking path. There is a major difference in these two approaches. In the former, the gait kinematics are measured from a rear view of the person, and in the latter, the gait kinematics are measured from a frontal view of the person. Regardless of the algorithm employed, because of the distance between the robot and the patient, the “follower robots” allow a complete view of the human body, making it possible to evaluate the gait kinematics of the full body.

Among the walker robots for gait analysis, the UFES Smart Walker developed by Cifuentes et al. [17,18] can be mentioned. This robot is used for human locomotion assistance. The platform detects navigation intentions of patients from a sensor fusion method that combines: a laser range finder (LRF) sensor to estimate the kinematics of lower limbs; wearable inertial measurement units (IMUs) to capture human and robot orientations; two triaxial force sensors to measure the physical interactions between the upper limbs and the robotic platform. Due to its low speed (up to 0.5 m/s), short following distance (0.5 meters), and the sensors it has for the analysis of gait parameters (LRF and IMUs), it can be considered as a walker robot.

Another interesting robotic platform is the ISR-AIWALKER presented by Paulo et al. [19]. The ISR-AIWALKER is a robotic walker for intuitive and safe mobility assistance and gait analysis. The platform applies a multimodal sensor structure that is capable of monitoring the user’s gait in close proximity. Using the RGB and depth map data, a kinematic model of the user’s lower limbs is obtained that allows the identification of a set of features that are used in a machine learning approach to discriminate gait asymmetries. For gait analysis, this robotic walker uses depth data provided by a Leap Motion in addition to a depth sensor (Intel F200). The use of several sensors is required due to the proximity of the user. The Leap Motion controller points to the center of the legs. Its purpose is to capture the motion of the waist and legs, allowing kinematic modeling of the lower limbs. Foot and ankle data are captured by the Intel camera. It allows heel-strike detection and extraction of other features for gait analysis.

Regarding the follower robots, the ROGER project [20] developed a mobile Socially Assistive Robot (SAR) to support patients after surgery in hip endoprosthetics. ROGER is a robotic gait coach that can navigate in clinic hallways, accompanying and observing the self-training of patients. By recognizing gait deviations, the robot is able to give corrective feedback immediately in the conditions of a real clinical environment. This robotic gait coach is planned to assist patients who have just received medical consent to walk with crutches. The robot has a height of 1.5 m and a maximum driving speed of up to 0.9 m/s. The localization system is based on an adaptive Monte Carlo approach, and detecting obstacles is performed through an occupancy grid mapping approach. To guarantee the best conditions for estimating the skeleton of the patient with the Kinect V2 during training, the patient is kept at an optimum distance and angle relative to the depth sensor. The robot detects a subset of gait characteristics, including step length, stance duration, step width, trunk lean, flexion/extension of knee and hip joints, and the crutch position. Since the Kinect V2 has issues with stable tracking of the foot positions, ROGER uses the ankle positions during the detection of gait phases due to their more accurate estimation.

The Lucia human-interactive medical robot developed by Ryo Saegusa [21] is another interesting approach that supports gait training based on autonomous evaluation and navigation. An infrared camera, ASUS Xtion, is mounted on the robot to develop the skeleton-tracking task. The robot assesses gait patterns of patients and guides them in gait training with auditory, visual, and somatosensory stimulation. Moving visual projections, sound generation, and local body vibrations intuitively inform subjects of the status of their own movements and the next target in the middle of motor execution. During the interaction, the robot autonomously controls the locomotion and the sensory stimulation. Unlike other approaches, this robot uses a trajectory previously taught to it by a physiotherapist. For this purpose, a force-sensing handle on the robot head is available. Once the robot has learned the trajectory, it can navigate autonomously while guiding the training subject on the previously learned trajectory.

Finally, the mobile robot presented by Bonnet et al. [22] was designed for pathological gait analysis from the rear view of the patient. This platform is based on the Pionner 3-DX mobile robotic platform. A Kinect sensor is mounted horizontally on a mast at a tunable height that is selected to be approximately at the pelvis height. This mast is adapted with a servomotor. Using a Kinect sensor, the robot is able to follow a patient at a constant distance on its own defined trajectory and estimate the spatio-temporal parameters by analyzing the gait from the rear view of the patient. The Microsoft SDK was not used for skeletal tracking. In contrast, this robot requires the placement of flat markers on the patient to assess the gait. Once the markers are found in the color image, the associated position in this image is calculated using the Robust Tracking–Learning–Detection (TLD) algorithm. The coordinates of the patient’s position in the color image provided by the TLD algorithm were used in combination with the calibrated depth map to obtain the 3D patient position in the Kinect sensor system of the reference. The algorithm estimates the positions of the targets attached to the trunk and heels of the patient. To conclude this review, Table 1 presents a summary of the robotic platforms mentioned above and others found in the state of the art. As can be seen in Table 1, depth sensors are commonly integrated into mobile robotic platforms. Among them, the most popular is the Kinect sensor. This is because the Microsoft Kinect SDK 2.0 is free and easy to use. On the other hand, the use of sensors such as Leap Motion and LRF has been proposed for gait analysis. However, their implementation is observed more in robotic walkers.

As a contribution to the state of the art, the idea presented in this study provides an innovative person-tracking strategy by using a follower robot equipped with a depth sensor. This strategy is especially designed for a gait analysis application because gait data are retrieved from the frontal part of the participant, in contrast with the current approaches. In addition, the mobile robotic platform has been tested in real environments, and the analysis of the applied configurations for navigation and person following was always compared with the quality of the joint kinematic signals.

## 3. Materials and Methods

The purpose of the robotic platform presented in this study is to be used in homes, hospitals, or clinics. To achieve this, the robot has to be able to navigate autonomously, avoid obstacles, and locate itself accurately on a map while following the subject and analyzing gait in a proper way. In the system presented in this study, these requirements were implemented using simultaneous localization and mapping (SLAM), control techniques, and path-planning algorithms. In this way, the evaluator defines the navigation goal on the map, after which the trajectory for the walking experiment is generated. During the experiment, the execution of the control tasks is performed by a double controller: lane keeping and person following. The following distance is flexible; however, the system is preconfigured for tracking the patient at 2.0 m, which was determined as optimal during the experimental stage of this study. An RGBD camera uses the Nuitrack SDK to perform the skeleton-tracking application by retrieving the three-dimensional positions of 19 joints of interest. The robot–person–environment interactions in a typical setup in the development of this study are presented in Figure 1. It is important to note that the limitations of the study are the fact that dynamic obstacle avoidance is not analyzed and that the laser sensor used for mapping and navigation can be replaced by another one, since glass walls are not detected. The following sections explain the mechanical design of the robot and cover the design of the controllers implemented in the system.

### 3.1. Mechanical Design and Components

The mechanical structure of the robot consists of eight components, which are labeled in Figure 2. At the lower end, there is a mobile platform (1) that provides the motor functions and allows the robot to move precisely. This mobile base is equipped with a CORE2-ROS controller and is in charge of managing the robot’s firmware. The mobile base is connected to an external computer, an Intel NUC (2), which resides in aluminum cabinets and is in charge of managing the wireless communication between the two computers through a Wi-Fi network. It is also in charge of the strongest and most resource-demanding jobs, the navigation and the motion capture using the depth sensor.

A LIDAR sensor sits on top of the cabinets, providing the robot with the ability to detect obstacles in the environment and navigate safely around them (3). Additionally at the top of the CPU cabinet is a set of adjustable aluminum brackets that regulate the height of the motion capture system (4). At the top of the robot, there is a mechanism that is specially designed for the servoing system (5). This is a two-piece rotary joint that operates similarly to a bearing. It is used to allow rotation of the head of the robot around the vertical axis and achieve a better fit. In the servoing system, the top piece moves over the bottom piece via a thin rail. This increases the friction very slightly, but greatly reduces the complexity of the servoing mechanism. The rotation capability of head of the robot is limited by the rotation range of the servo motor (6), which ranges from 0 to 180 degrees. In addition, due to the height of the robot, it can easily be exposed to oscillations and vibrations that affect the data acquisition. For this reason, the integration of a damping device (7) is necessary. Finally, an Orbbec Astra RGB-D camera is located in the head of the robot, which is responsible for the skeleton-tracking task (8).

### 3.2. Navigation and Mapping during the Walking Experiment

ROBOGait is an open-source autonomous robotic platform based on the Robot Operating System (ROS). It is equipped with the ROS Melodic distribution and the Ubuntu 18.04 (Bionic) release. Processes in ROS are executed with a node-based approach. The implemented ROS architecture and data flow in the mobile robotic platform are presented in Figure 3. As can be appreciated in this figure, the mobile robotic system has six nodes of special relevance, which are described as follows:Slam_gmapping or map_server nodes. The gmapping node publishes the/map topic containing the navigation map. In the case of the use of a previously built map, the map_server node is executed instead. The functions of both nodes are similar.Move base node/teb_local_planner. The move_base node provides the global plan for navigation, while the teb_local_planner node works as a plug-in for the ROS move_base package and is responsible for providing the local plans for navigation (those used for obstacle avoidance).Firmware node. The firmware node publishes the status of the motors, sensors, and odometry of the robot.RViz node. This is used for visualization and as an interface for sending navigation goals.Nuitrack body tracker node. This node is responsible for skeleton tracking using the Nuitrack SDK.Person follower and Lane Keeping node. This node executes two control tasks. It is responsible for keeping the robot–person distance and for maintaining the trajectory during navigation.

As mentioned above, to build the map and localize the robot on it, the slam_gmapping node of the gmapping package is used. This package provides simultaneous mapping and localization (SLAM) using the laser sensor data from the robot. To do this, it takes the sensor_msgs/LaserScan messages and builds a map that is published in a /map topic with the nav_msgs/OccupancyGrid message type.

Once the mapping is complete, the system is ready to perform autonomous localization and path planning. The task of path planning for a mobile robot consists of determining the sequence of maneuvers that the robot must perform to move from the starting point to the destination while avoiding collisions with obstacles. In the ROS, it is possible to plan a trajectory based on occupancy grid mapping, i.e., using the messages obtained from the slam_gmapping node. This planner is a node named move_base, which is from the ROS move_base package. Although this planner provides a global and local plan for navigation, there are other local planners that can provide better results and can be added as plug-ins for the move_base node. As mentioned above, in this study, the teb_local_planner has been used as a plug-in for local planning. The local plan is essential in a system that is used in a real environment because it allows replanning of the trajectory during execution, thus providing the system with the ability to avoid unexpected obstacles. The teb_local_planner implements the Timed Elastic Band (TEB) algorithm proposed in [27,28]. This algorithm locally optimizes the robot’s trajectory with respect to execution time, obstacle avoidance, and compliance with the kinematic constraints of the robot.

In summary, once a navigation goal is set on the map, the move_base node provides a global navigation plan and the teb_local_planner provides the local plan for obstacle avoidance.

In addition, the gait patterns are quite different between different affectations. There are people who can walk faster, as in the case of MS patients, and others who walk more slowly, as in CP or stroke patients. Therefore, free speed of the robot cannot be allowed while following the navigation plan. The robot must match its speed to the person’s walking speed. This ensures not only the use of the system for the analysis of different pathologies, but also a better data capture, since the capture distance remains constant; nor can the robot be allowed to alter the global trajectory too much to avoid an obstacle, because extreme situations can arise that spoil the experiments. In this context, it is necessary to establish a series of strategies and boundaries that allow navigation to be coupled to a gait experiment. In this study, the behavior of the platform was adapted to these conditions:Two operating zones are defined according to the lateral deviation (LD) from the given global plan: the Safe Zone and Reinsertion Zone (Figure 4). In the Safe Zone, the local plan provides the steering points for driving the robot. In the Reinsertion Zone, the global plan is the one that provides the steering points to drive the robot.Considering that the robot is used in walking experiments and not for free navigation in an environment, obstacle avoidance must be limited to prevent the robot from totally deviating from the global plan. The limit of the Safe Zone was established to be 0.70 m on each side of the global navigation. This is the area where the robot is allowed to avoid obstacles. This zone was set to fit with the size of medium obstacles, such as trash cans, chairs, or the presence of a person. Within the Safe Zone, a steering point of the local plan is used. This corresponds to the third point of the navigation plan generated by the local planner. In that sense, in the presence of an obstacle, the robot follows the local plan to execute the avoidance task, but this task is limited to be executed only within the Safe Zone. This prevents spoiling of the experiment by an excessive deviation of the robot. The lower image of Figure 4 illustrates how this controller condition works.Outside the Safe Zone is the Reinsertion Zone. In this zone, the robot is driven using steering points calculated with data from the global plan (not from the local plan). This is a zone that is more adaptable to the system dynamics and velocity changes during the experiment. Within the Reinsertion Zone, the robot defines a trajectory that allows it to reinsert itself into the Safe Zone. Once the robot has entered into the Safe Zone, the robot continues with the lane-keeping task, following the global plan in the normal way. The limit of the Reinsertion Zone was 1.40 m from the global plan. This is a value that was established experimentally. If the distance from the robot to the path is greater than this value, the system executes an emergency stop. The upper image of Figure 4 illustrates how this controller condition works.Regarding the tracking of the person during the navigation, while an obstacle avoidance task is executed, or even at any instant of the walking experiment, the robot may lose the person. In this case, the “confidence” parameter from the Nuitrack SDK is used. This parameter is used to quantify the quality of the skeleton tracking. Any joint confidence below a certain value will be too noisy or unavailable. Any joint confidence above this certain value will be good enough to be used. This confidence level gives feedback on whether the person is well tracked. In the algorithm implemented in ROBOGait, if the confidence is lower than 0.5, the controllers implemented in the robot continue driving the robot by making use of the last well-captured value from the skeleton. This gives the robot the opportunity to recover the tracking of the person. If one second of time has passed and the person still has a low confidence in any of the joints, the experiment is aborted and the robot is stopped for security.

### 3.3. Design of Controllers

Regarding the design of the robot motion controllers, during the walking experiment, the linear velocity of the robot is provided by a person-follower control loop that maintains the relative robot-to-person distance. The angular velocity is provided by a lane-keeping control loop. Figure 5 shows the control scheme implemented in the mobile robotic system. The person-follower controller was already explained in a preliminary study presented with this platform in [29], where an ROS-based straight-line follower robot for human gait analysis was designed. The reader is encouraged to review this document for more detailed information on this controller. Therefore, this section will focus on the implementation of the lane-keeping controller. In addition, the design of the servoing system controller that ensures that the camera is always pointing in the direction of the tracked person is shown. These controllers have an important role in achieving successful walking experiments.

#### 3.3.1. Lane-Keeping Controller

A lane-keeping control law was developed. This law allows the robot to follow the planned trajectory during the walking experiment. Figure 6 gives an overview of the proposed strategy.

The target for the lane-keeping controller corresponds to the angle that the robot must rotate to be aligned with the steering point. This target angle $\theta_{T_{\left(FM\right)}}$ was calculated as follows:(1)$$\theta_{T_{\left(FM\right)}}=\theta_{s_{\left(FM\right)}}-\theta_{r_{\left(FM\right)}}$$
where $\theta_{r_{\left(FM\right)}}$ represents the angular position of the robot in the map frame, and $\theta_{s_{\left(FM\right)}}$ represents the angular position of the steering point. The steering point corresponds to the third point of the navigation plan generated by the local planner, i.e., the TEB planner. The first point of the local plan is not used because it is very close to the robot and the angular difference between the robot and this steering point is minimal.

The output for the control law represents the angular speed of the robot ${\dot{\theta }}_{r}$. The first approach to computing the speed command was to implement a proportional controller as follows:(2)$${\dot{\theta }}_{r}=K_{p}\;\theta_{T_{\left(FM\right)}}=K_{p}\left(\theta_{s_{\left(FM\right)}}-\theta_{r_{\left(FM\right)}}\right)$$

Considering the rotational model of the robot as an integrative process, the controller gain $K_{p}$ can be estimated according to the following equations:(3)$$G_{c}\left(s\right)=K_{p};G_{p}\left(s\right)=\frac{1}{s}$$
(4)$$G_{cc}\left(s\right)=\frac{G_{c}\left(s\right)G_{p}\left(s\right)}{1+G_{c}\left(s\right)G_{p}\left(s\right)}=\frac{1}{(\frac{1}{K_{p}})s+1}=\frac{1}{Ts+1};$$
where $T=\frac{1}{K_{p}}$ is the time constant of the closed-loop system. A widely used criterion for choosing the value of the constant T is that it should be at least 10 times the controller sampling time. In our application, the controller frequency is 30 Hz, and therefore, the controller sampling time is $T_{c}=33.3$ ms. With this, the gain for the proportional controller is:(5)$$K_{p}=\frac{1}{T}=\frac{1}{10T_{c}}=\frac{1}{10(33.3\mathrm{ms})}=3$$

In order to evaluate the effectiveness of this controller, an experimental stage was carried out. Using ROS tools, a series of steering points were emulated. These points emulate the steering angles arriving from the local planner. At the same time, the response of the robot to reaching these targets was measured using an inertial sensor, the MPU-9250.

The results of this experimental stage are shown in Figure 7. The figure shows seven set points sent sequentially: ±0.52, ±0.35, ±0.17, and 0 rad. Figure 7 shows how the controller corrects the steering angle of the robot. When the robot rotates from +0.52 to −0.52 rad, it performs an angular rotation of 1.04 rad, which is one of the highest values of steering angle expected during the lane-keeping task. To make this correction, the control signal generated is approximately 3 rad/s. This value is far less than the limit of the angular velocity of the robot (7.33 rad/s). This means that the controller gain can be further increased to achieve a faster correction; however, the angular acceleration of the robot prevents us from forcing the control action excessively, since the system can become unstable, i.e., once the robot reaches the set point, it may not stop, which would cause the robot to start spinning. An attempt was made to incorporate a derivative action in the controller to make a speed correction, but the results were very similar.

Further experiments will be described in the experimental section of this study to evaluate the error of the lane-keeping controller on a real trajectory.

#### 3.3.2. Servoing System Controller

The purpose of the servoing system controller is to ensure that the camera is always pointing in the direction of the tracked person. For this purpose, what is controlled is the angular position of the person in the camera frame $\theta_{p_{\left(FC\right)}}$. This angle can be observed in Figure 6. The spine base coordinates retrieved from the skeleton-tracking application are used to calculate the angle $\theta_{p_{\left(FC\right)}}$ as follows:(6)$$\theta_{p_{\left(FC\right)}}=tan^{-1}\left(\frac{Y_{person_{\left(FC\right)}}}{X_{person_{\left(FC\right)}}}\right)$$

The first approach to computing the control law was to implement a proportional controller by using the angular position of the person in the camera frame $\theta_{p_{\left(FC\right)}}$ as the actual measurement and a 0° angle as the target. The output of the control law is the final angular velocity of the servomotor calculated on the basis of the person’s deviation and the controller’s cycle time, as follows:(7)$${\dot{\theta }}_{camera}=K_{p}\left(\theta_{p_{\left(FC\right)}}\right)=K_{p}\left(\theta_{p\_target_{\left(FC\right)}}-\theta_{p_{\left(FC\right)}}\right)$$

It is important to mention the constraints of each of the parameters related in this control law. The maximum rotational speed of the servo motor Hitec HS-755HB is 3.74 rad/s; this is the saturation vale of this controller. On the other hand, the manufacturer of the Orbbec Astra sensor states a field of view (FOV) of the following magnitudes: (1.05 H × 0.86 V) radians. When analyzing the horizontal FOV, it can be deduced that the maximum angular deviation that the person can experience with respect to the camera frame is ±1.05 rad. This is the maximum error expected in the control law. If the person deviates from the camera frame more than this value, the camera loses the person, and thus, the skeleton-tracking application fails. These are important features that are used to adjust the controller. Therefore, it is possible to try that, at 1.05 rad, the controller uses the maximum speed of 3.74 rad/s. By relating these two values, the gain of the controller is: $K_{p}=3.56$.

The friction of the rotating mechanism, the weight of the damping system, and the weight of the camera can cause the servo not to reach the maximum speed of 3.74 rad/s. This prevents one from forcing the behavior of the controller too much. In this way, to set the value of the controller gain Kp, two alternatives were tested: Kp=1.5 and Kp=3.0.

The servomotor system is a fundamental part that influences the quality of the skeleton tracking. Therefore, the evaluation of which of the two controller gains gives better results will be further developed by evaluating it in conjunction with the quality of skeleton tracking.

### 3.4. System Validation

Nuitrack SDK was used as a solution for motion capture. The main objective of Nuitrack is to establish an API for communication with RGBD sensors. In this study, an Orbbec Astra sensor is used. Each Nuitrack skeleton has 19 joints. Each joint has a position and orientation.

Although this tool is very useful for capturing gait data, it is necessary to measure the error of the SDK when working together with the mobile robotic platform ROBOGait. For this purpose, several walking experiments were carried out. Thirty-seven participants—21 healthy men and 16 healthy women—with an average age of 21 ± 2 years, an average height of 172.99 ± 8.53 cm, and average weight of 67.41 ± 10.28 kg were involved in the study. In each experiment, joint trajectories were recorded with the robotic system at a sampling rate of 30 Hz. The accuracy of the system was measured by comparing it with a six-camera Vicon M2 MCAM system working at a sampling rate of 120 Hz. A total of 207 gait recordings were processed. This study was approved by the ethics committee of the Faculty of Physical Activity and Sport Sciences INEF (Polytechnic University of Madrid, Spain). The experiments were performed in the Sports Biomechanics Laboratory of INEF. As a summary, the experimental protocol involved two stages, which are described as follows:Preparation of the participant (15 min). To capture the kinematics of the whole body with the Vicon system, 32 reflective markers were placed on anatomical landmarks following the Plug-in-Gait model. The participant was asked to wear running shoes, short tights, a top (in the case of women), and socks below the ankles. Additionally, the participant was asked to sign an informed consent form and complete a survey with basic questions related to health problems and musculoskeletal injuries. Subsequently, their height and weight were measured. Before starting the test, a warm-up and normalization of gait speed to 1 m/s were performed.Walking tests (15 min). Participants walked in a 15 × 5 m corridor in a one-way straight-line gait, as shown in Figure 8. This exercise was repeated 10 times with each participant, but only the five or six best recordings from each participant were processed. At the end of this experimental stage, a total of 207 gait recordings were processed.

In order to compare both systems, some preliminary steps were necessary. Nuitrack provides a skeleton model from which the three-dimensional coordinates of various body landmarks can be obtained with the depth sensor. The first step is to design the inverse kinematics that will be applied to this skeleton model to obtain the joint angles during the gait. It should be attempted to make these inverse kinematics as similar as possible to those applied in conventional gait models. This is very difficult to achieve with the robot, since there are only two markers per body segment, as opposed to the conventional model, where at least three markers per segment are used. The initial task is to find an equivalence between the landmarks detected by both systems by defining certain mapping rules. These mapping rules help to relate as much as possible the inverse kinematics applied by the robot with those applied by the Vicon. Once the inverse kinematics are calculated with the robot system, the error with respect to the Vicon system can be determined. Each of these processes is explained in a preliminary study developed with the robot [29]. We invite the reader to review this paper for more information about the inverse kinematics process applied with the robot.

Table 2 shows the results of this validation for joint kinematics in the sagittal, frontal, and transverse planes. Figure 9 shows the comparison of kinematics detected by the systems. Only the kinematics that the robot detects most accurately are included in the analysis. When analyzing the correlations shown in Table 2, it is observed that almost all joint movements in the sagittal plane were well detected, given the principle of operation of the depth sensor, which is designed to detect movements in depth. The elbow, knee, hip, and shoulder presented correlation values higher than 0.7. This was different in the case of trunk and pelvis tilt, where the correlation was lower than 0.7. In addition, as can be observed in Figure 9 the trunk tilt and pelvis tilt were affected by an offset with respect to the signal of the Vicon system. These problems could arise from the difference in the skeletal models between the Vicon and robot systems. Since the number of markers detected by the robot was smaller than those detected by the Vicon, the definition of the coordinate systems of the body segments was, in fact, different in each system. While the Vicon uses four markers to detect the kinematics of the pelvis, the robot must use only two markers, plus one belonging to the spine. Despite this fact, considering the low range of motion of the trunk tilt and pelvis tilt (about two degrees), the results are promising. For the detection of motion in the frontal plane and transverse plane, the system relies on the intrinsic parameters of the RGB camera, so these motions are approximations of medium quality. However, correlations in these planes were higher than 0.7.

Gait can be described with the support of kinematic and spatio-temporal parameters. Some of these parameters can also be considered descriptors of gait disorders and are the key point in differentiating normal from pathological gaits. Therefore, it is very important to analyze them separately from the rest of the kinematic analysis described above. Table 3 summarizes the root-mean-square error of the robot with respect to the certified Vicon system when detecting the main descriptors of gait. This demonstrates the ability of our system to measure these parameters during a gait analysis.

## 4. Results

### 4.1. Evaluation in Real-World Environments

Before testing the system in a real clinical environment, preliminary tests were carried out in the corridors of the Universidad Politécnica de Madrid. These corridors resemble those of a clinic or hospital. These experiments were carried out to fine-tune the control systems. This adjustment was made on the basis of two criteria: the quality in following the planned trajectory and the quality in capturing the kinematic gait signals.

The map has combinations of straight lines and curves through which the platform moves while recording the gait data from the participant. Using ROS tools for data acquisition, the planned trajectory was collected, as well as the trajectories followed by the person and the robot during the gait experiment. For experiment 1, a servoing system controller gain of Kp=1.5 and a following distance of 2.5 m were set. Figure 10a shows the planned path and the real path followed by the robot and the person, as well as the kinematic signals of flexion/extension of the knee and hip during experiment 1. This figure contains timestamps that allow one to know the positions of the robot and the person and the corresponding kinematic signals along the trajectory. Additionally, this figure shows the angular position of the person in the camera coordinate system.

Experiment 1 lasted approximately 50 s, and a total of 26 consecutive gait cycles were recorded. The results of experiment 1 are shown in Figure 10a. It can be noted that in the straight sections of the trajectory of experiment 1, the kinematic signals were similar to the waveform described in the literature [30]. Unsurprisingly, in the curved sections, this was not the case. Focusing on the range between 24 and 33 s, where the robot was passing through a corner, it can be seen how the kinematic signal of knee flexion was affected. The signal was attenuated and sometimes deformed by both the change in the point of view of the camera and the maneuver itself. No affectations were observed in any other gait kinematic signals. When analyzing the deviation of the person with respect to the camera coordinate system during experiment 1, it can be easily inferred that, in this section, the correction performed by the servoing system was very slow. Consequently, the system allowed an excessive deviation of the person, and the effects caused by occlusion between the limbs and loss of camera focus are very noticeable.

Another variable that can affect capture in these sections is the following distance of the person. During this experimental stage, it was noticed that the greater the distance between the robot and the person, the more difficult it was to capture the skeleton in a curved section. This is because, in corners, the relative rotation increases, and the camera practically observes the person standing sideways. During the experiment, it was observed that there is even a high risk of the robot losing the person because the wall blocks the line of sight.

Based on the results obtained in experiment 1, it was decided to make changes in the experimental setup. The purpose of these modifications was to achieve improvements in the signals in curved sections (specifically in the corners).

These changes focused on the servoing system and the following distance. An increase in the gain value of the servoing controller could achieve a faster correction of the angular position of the person in the camera coordinate system. Therefore, the gain value was changed from Kp=1.5 to Kp=3.0.

Regarding the following distance, it was previously set to 2.5 m. This parameter was slightly reduced at 2.0 m to keep the robot closer to the participant. This helped to prevent the camera from looking sideways at the participant in corners.

Experiment 2 was conducted on the basis of these changes, and the results are shown in Figure 10b. The experiment was performed with the same participant and in the same environment as experiment 1. Experiment 2 lasted approximately 49 s, and a total of 26 consecutive gait cycles were recorded.

When analyzing the kinematic signals obtained in experiment 2 between 23 and 31 s (when the robot was moving around the corner), the difference with respect to experiment 1 can be noticed. The illustration of the angular deviation of the person with respect to the camera coordinate system shown in experiment 2 suggests improved person tracking. As a result, the kinematic signal of knee flexion/extension had an improved waveform. It can also be noted that the hip flexion/extension had some attenuation, but its levels were still within acceptable values.

Regarding the quality in following the planned trajectory, the global plan data were compared with the real trajectory followed by the robot during the experiments. The comparison between these trajectories was made somewhat difficult because the planned trajectory was a vector with 2D coordinates, while the trajectory followed by the robot was reported as a time series. It was then decided to make the comparison based on the position of the coordinate point on the map. That is, given a position of the robot on the map, it was compared with the nearest coordinate of the planned path vector. This is a fair and quite realistic comparison. The comparison was performed using the Euclidean distance, and results are shown in Table 4.

The results were satisfactory; the average deviation distance from the path was less than 0.05 m. As observed in Figure 10, in both experiments, the maximum deviation of the robot from the global plan occurred just after the robot moved around the corner. This maximum deviation was less than 0.20 m. This means that the lane-keeping controller worked properly and kept the robot within the Safe Zone.

To conclude the design of the controllers and after the experimental results were obtained, it was decided to set the controller values as follows: $Kp_{LaneKeeping}=3.0$, $Kp_{Servoing}=3.0$, and a robot-to-person following distance of 2.0 m.

### 4.2. Experiments in Clinical Environments with MS Patients

During the experiments, patients with relapsing–remitting multiple sclerosis (RRMS) underwent gait tests. These tests were not intended to analyze the functional status of the patients, nor the progress of the disease or the deviation of the gait pattern from normal. These topics will be discussed in a future work. The purpose of these tests was to analyze the effectiveness of the lane-keeping controller and to extract the gait patterns from the patients to verify that the resulting waveforms correspond to what is described in the literature.

The only variable controlled during the experiment was the interaction with dynamic obstacles. The environments in which the tests were conducted presented a higher degree of complexity than the environment used for the preliminary tests. Doors, glass walls, and narrow corridors found in these clinical environments made testing a challenge. The clinical environment “A” was the Getafe Multiple Sclerosis Association, and the clinical environment “B” was the University Clinic of the Rey Juan Carlos University. Both buildings were indoor environments with artificial lighting. During the experiments in real corridors with MS patients, the only variable controlled was the interaction with dynamic obstacles.

Figure 11 shows the results of this experimental stage. In the upper part of Figure 11, the planned path and the real path followed by the robot and the MS patient are shown. The images in the lower part of Figure 11 show the averaged and normalized (0–100%) gait patterns obtained for knee and hip flexion/extension. As can be seen, the waveforms resulting from the experiments are consistent with those described in the study of [30].

Regarding the lane-keeping task, Table 5 summarizes the robot’s deviation from global plan during the experiments. When looking at the dimensions of the planned paths, it can be seen that the path in environment “A” was shorter than the rest. This is because this clinical environment is smaller than the other environments. Navigation in narrow spaces was highly difficult because the robot needed a higher reaction speed to be able to execute fast corrections of its trajectory. This is why the robot’s deviation from the path in environment A was slightly higher than in environments “B1” and “B2”. Finally, since the experiments performed in this stage were conducted in more complex environments, the maximum robot deviation (which, in one case, reached 0.3193 m) was considerably higher than in the preliminary experiments described in the former section (where the maximum deviation was 0.1837 m). However, in none of the cases did the robot leave the safe navigation zone.

## 5. Discussion

The use of mobile robots for human gait monitoring and analysis is a field generating a great deal of interest in the medical community. All studies focused on this topic are looking for alternatives to expensive photogrammetric systems. The purpose of this section is to discuss the differences between our mobile robotic system with the most current approaches.

In the robotic system proposed in this study, the error in the detection of gait kinematics is low. Differences in the gait model applied by the robot and the Vicon system affect the correlation at certain joints; however, the results of our system are promising when compared to those presented in the literature. If we compare our system with the mobile robot presented by Bonnet et al. [22], where gait analysis from the rear view of the patient was performed, the error rates reported in the research of Bonnet et al. (8° knee flex/ext., 4° hip flex/ext., and 2° hip add/abd.) are slightly lower than those that we report in our study (5.94° knee flex/ext., 4.52° hip flex/ext., and 2.59° hip add/abd.). Nevertheless, it is worth mentioning that the results reported by Bonnet et al. were obtained with the subject walking on a treadmill. On the contrary, in our study, the results were obtained from real overground walking. Despite this fact, the Pearson correlations reported in our study (0.95 knee flex/ext., 0.95 hip flex/ext., 0.70 hip add/abd.) were slightly higher than those reported by Bonnet et al. (0.88 knee flex/ext., 0.88 hip flex/ext., 0.65 hip add/abd.).

The Xanthi Rollator Walker presented by Xanthi Papageorgiou et al. [23] is a completely non-invasive approach that uses a non-wearable device for human pathological gait analysis. The system is based on a typical LRF sensor that is used to detect the lower limbs. The experimental results with this platform demonstrated an error of 0.0278 m for stride length, 0.0362 s for stride time, and 0.0418 s for swing time. These values are slightly lower than those of our platform (0.05 right stride length, 0.06 s for right stride time), but in the experiments with this platform, the participant walked with physical support of the rollator, which greatly increased the accuracy of the sensor, since the movements are very smooth. On the contrary, in our follower robot, the normal walking speed imposed by the patient had preference. Since there is a predefined distance between the robot and the person, it is possible to analyze the kinematics of the whole body, which offers greater possibilities for detecting anomalies.

Regarding the person-following algorithm and path-following algorithm, which involve two tasks—person tracking and safe robot navigation—it is interesting to discuss the mobile robotic platform presented by Doisy et al. [24]. In this study, two person-following algorithms that use depth images from a Microsoft Kinect sensor for person tracking are proposed. The first one, the path-following algorithm, reproduces the path of the person in the environment. The second one, the adaptive algorithm, uses, in addition, a laser range finder for localization and dynamically generates the robot’s path inside a pre-mapped environment, taking into account the obstacles’ locations. In both algorithms, the robot captures the gait from the back of the participant. The overall results of the evaluation of the trajectory-tracking algorithm on the robotic platform of Doisy et al. demonstrated a path error of 0.11 meters. In our system, where the person is followed from the front, 0.04 m of path deviation was obtained for experiments with healthy participants and 0.08 m for the experiments with MS patients.

In the study presented by Cifuentes et al. [18], a robot for walking assistance and analysis was developed. The author performed two types of experiments to evaluate the system. The first experiment executed an S-shaped trajectory with the robot working without traction. This means that the human guided the robot with their arms. The second experiment performed the same S-shaped trajectory, but the robot followed the human without physical interaction. The main concern when analyzing the study of [18] is that the tests were performed at two walking speeds, 0.25 and 0.5 m/s, for the validation of the proposed methodology. The user claimed that the selected speeds were consistent with the slower gait typically performed by people with disabilities. This is fully refutable, since the gait speed in people with MS has been measured to be 1.65 ± 0.34 m/s [31]. On the contrary, our platform has a maximum speed of 1.25 m/s. This is at least enough to replicate normal gait in patients with relapsing–remitting multiple sclerosis (RRMS), as was demonstrated during the experiments with MS patients presented in this study. This maximum speed would also be appropriate for self-selected speed capture in patients with secondary progressive multiple sclerosis (SPMS) and primary progressive multiple sclerosis (PPMS).

The system presented in this study demonstrated its ability for gait objectification and its applicability in clinical settings. Clinical tests with patients with MS gave an initial impression of the applicability of the instrument in patients with abnormal gait patterns. The idea presented in this study provides an innovative person-tracking strategy that is specially designed for gait analysis, since the data are obtained from the frontal part of the participant, which is an important contribution to clinical practice.

## 6. Conclusions

In this study, we analyzed the viability of using a mobile robotic platform for human gait analysis and established the basis for the design of a new prototype based on depth cameras to perform the analysis of human gait in practical scenarios. The purpose of this study was to provide the validity of this robot and its applicability in clinical settings. The mechanical and software design of the system was presented, as well as the design of the controllers for lane keeping, person following, and the servoing system.

During the validation procedure, the accuracy of the system in retrieving kinematic gait data and the main descriptors of gait was calculated with respect to the ground truth of a Vicon system. For this purpose, two hundred and seven gait recordings were processed thanks to the collaboration of thirty-seven participants. The results demonstrated high correlation and low error rates, especially in joint excursions from the sagittal plane. This was different in the case of trunk and pelvis tilt, where the correlation was lower than 0.7. This was attributed to the difference in the skeleton models between the Vicon and robot systems, since the number of markers detected by the robot was smaller than the number detected by the Vicon.

Once the validation of the system was carried out, the tests were dedicated to evaluating the accuracy of the lane-keeping controller and the quality of the skeleton capture system in real environments. This experimental stage revealed that in curved sections, especially in the corners of a map, the knee joint is affected by occlusion and the deviation of the person in the camera reference system. This problem was greatly improved by adjusting the servoing system and the following distance to 2 m.

Once the system was adjusted, clinical tests with patients with MS gave an initial impression of the applicability of the instrument in patients with abnormal walking patterns. These tests reveal the advantages of using the proposed system in clinical settings, allowing the analysis of gait in patients outside the laboratory environment, without the limitation of walking space and without the need for long preparation times.

As a distinction from current approaches, where there is an almost total tendency for mobile robotic systems to follow the person and capture motion from the rear, in this study, the robot maintains the distance from the person while following a predetermined trajectory by capturing the gait from the frontal side of the participant. This strategy was designed specifically for human gait analysis and is also a contribution of this study to clinical practice.

It is important to note that the limitations of the study are the fact that dynamic obstacle avoidance was not analyzed and that the laser sensor used for mapping and navigation can be replaced by another one, since glass walls are not detected. Finally, further analysis is required to identify the differences of pathological patterns with respect to normal ones by using data acquired by the proposed robotic system.

Overall, this study describes the entire path from the idea to an actually implementable instrument, whereby conditions regarding clinical applicability and accuracy were constantly tested. The idea of using depth cameras in combination with a mobile robot system for the objectification of gait is interesting and offers new possibilities for the user-friendly and non-invasive performance of gait analysis in clinical practice.

## Figures and Tables

**Figure 1 sensors-21-06786-f001:**
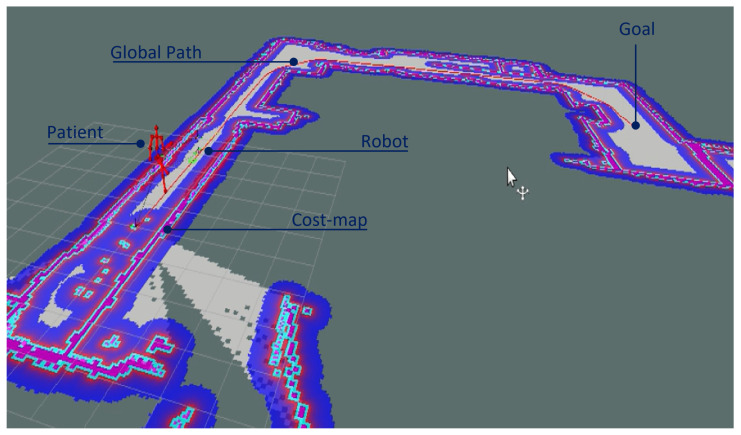
Robot–person–environment interactions in a typical setup in the development of this study.

**Figure 2 sensors-21-06786-f002:**
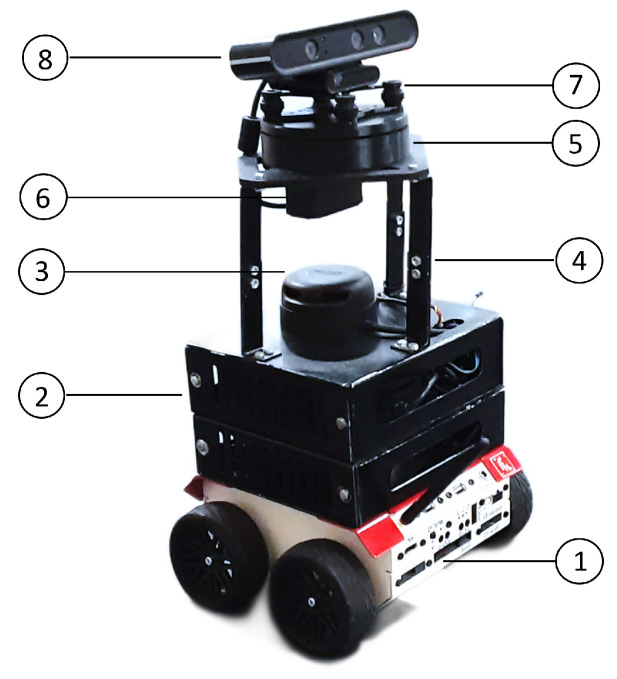
An overview of the mechanical design of ROBOGait. The figure shows: (1) the mobile robotic base, (2) on-board computer (Intel NUC), (3) Slamtec LIDAR A2 sensor, (4) adjustable aluminum brackets, (5) servoing system mechanism, (6) servomotor (Hitec HS-755HB), (7) damping device, and (8) Orbbec Astra RGBD camera.

**Figure 3 sensors-21-06786-f003:**
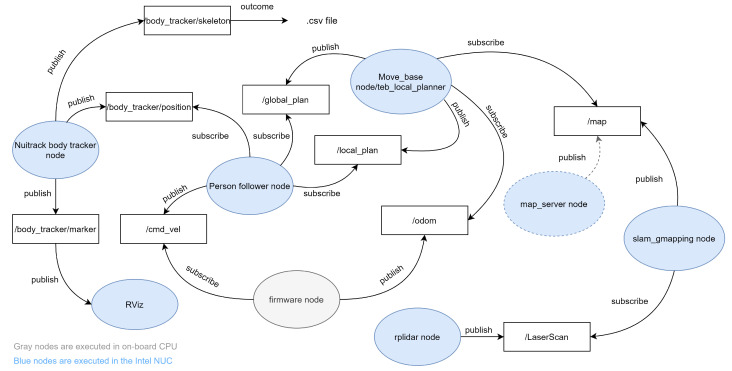
Data flow in the mobile robotic platform following the ROS publisher/subscriber communication protocol.

**Figure 4 sensors-21-06786-f004:**
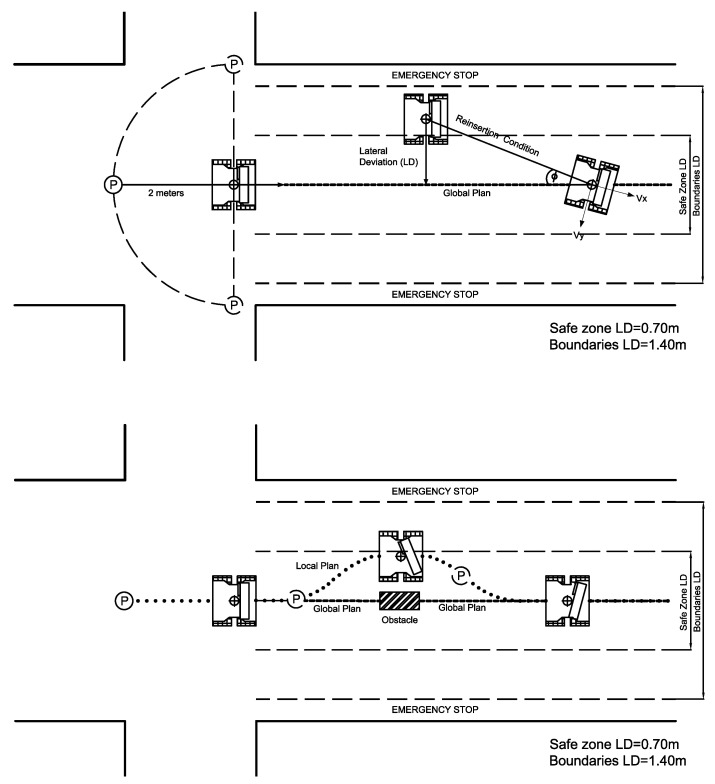
Schematic diagram of the operation of the lane-keeping controller.

**Figure 5 sensors-21-06786-f005:**
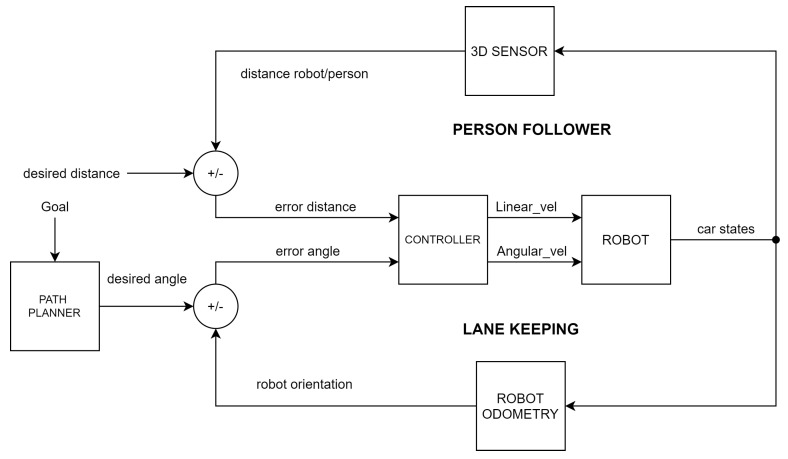
Person-tracking and lane-keeping controller.

**Figure 6 sensors-21-06786-f006:**
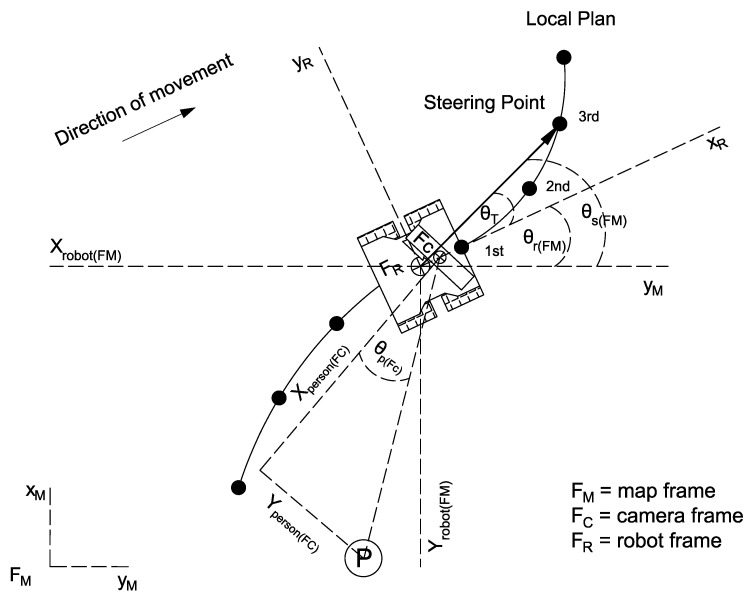
Points of interest for the calculation of the desired angle during the lane-keeping task.

**Figure 7 sensors-21-06786-f007:**
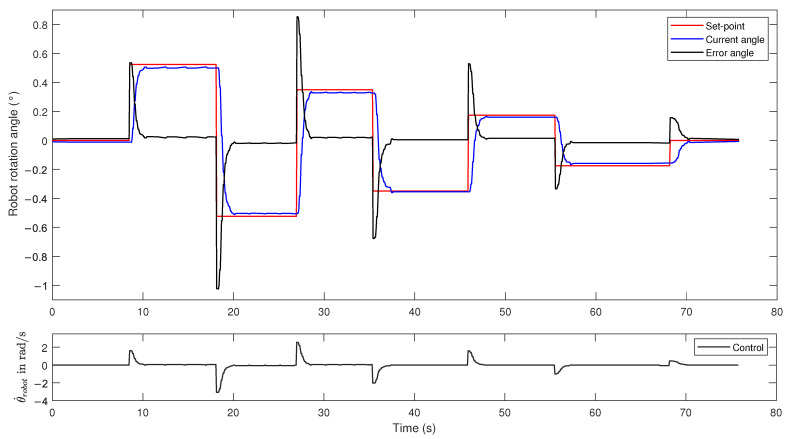
Robot’s behavior with the lane-keeping controller. The figure shows seven set points emulated sequentially: ±0.52, ±0.35, ±0.17, and 0 rad.

**Figure 8 sensors-21-06786-f008:**
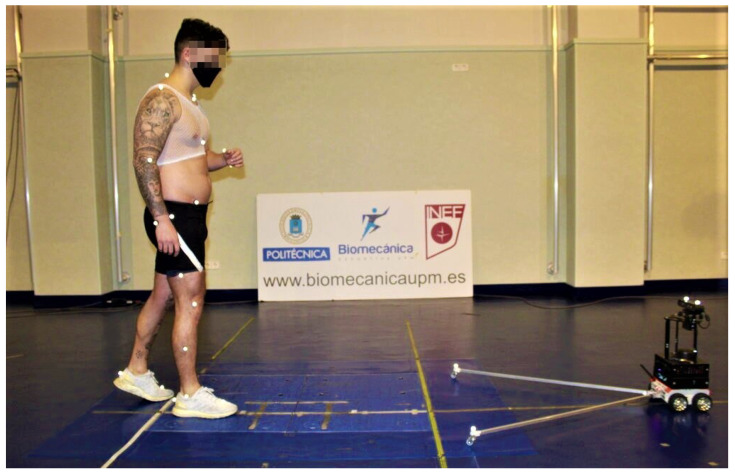
An overview of the experimental environment during the validation stage.

**Figure 9 sensors-21-06786-f009:**
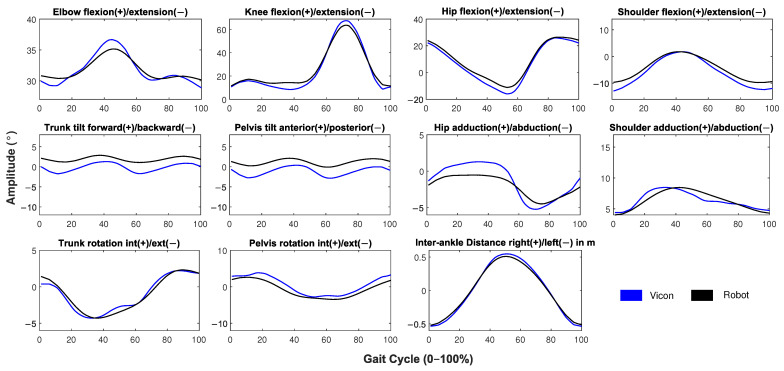
Comparison of kinematic gait cycles retrieved by the robot system (black line) and the Vicon system (blue line). The gait cycles are normalized from 0–100% and averaged for all of the iterations.

**Figure 10 sensors-21-06786-f010:**
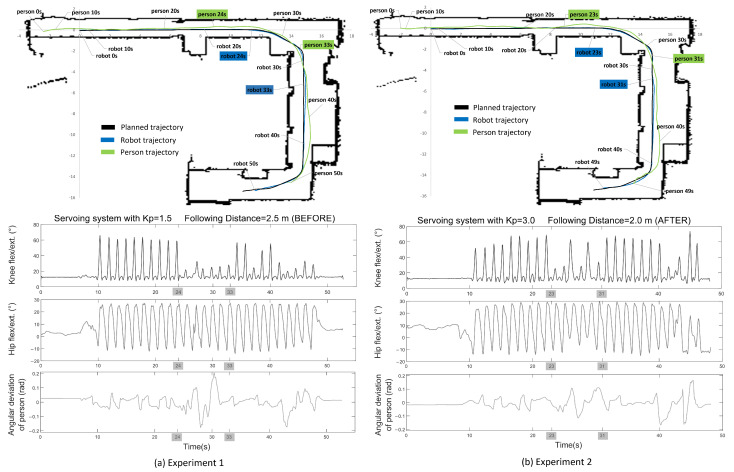
Experiment 1. Servoing system gain of Kp=1.5 and following distance of 2.5 m. Experiment 2. Servoing system gain of Kp=3.0 and following distance of 2.0 m. The maps show the planned path and the real path followed by the robot and the person. The kinematic joint signals (knee and hip flex./ext. in the entire experiment) in addition to the angular position of the person in the camera coordinate system.

**Figure 11 sensors-21-06786-f011:**
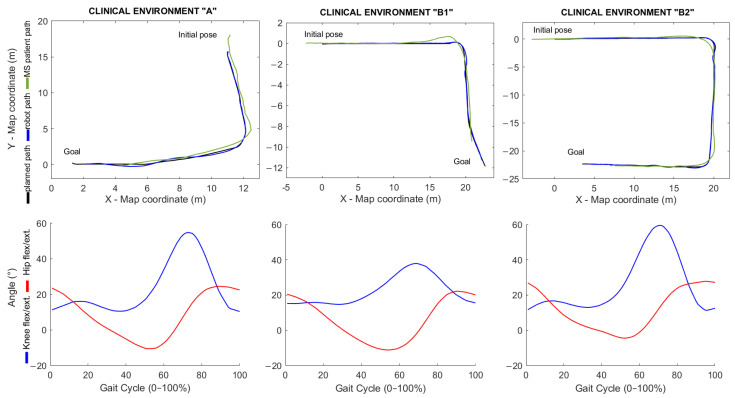
Walking experiments with MS patients in clinical environments. The upper part shows the trajectories, and the lower part shows the kinematic signals of knee and hip flexion/extension.

**Table 1 sensors-21-06786-t001:** Summary of robotic platforms for human gait analysis.

Robotic Platform	Max. Speed (m/s)	HRI	Mobile Base	Sensor for Gait Analysis
ROGER [20]	0.9	cHRI	SCITOS	Kinect V2
LUCIA [21]	-	cHRI	-	LRF, ASUS Xtion
UFES [17]	0.5	pHRI	-	LRF, IMUs, Triaxial force sensors
XANTHI [23]	-	pHRI	-	LRF
BONNET [22]	1	cHRI	Pioneer 3-DX	Kinect, Servomotor, Flat markers
ROBULAB10 [24]	-	cHRI	RobuLAB10	Kinect, LRF, Pan-tilt mechanism
HUANGHE [25]	1	cHRI	Pioneer 3-DX	Kinect v2, instrumented insoles
ISR-AIWALKER [19]	-	pHRI	-	Leap Motions, Intel F200
i-ROBOT [26]	0.5	cHRI	iRobot Create	Kinect

**Table 2 sensors-21-06786-t002:** Results of the validation of the robotic system in comparison to a Vicon system. The table shows the root-mean-square error (RMSE) and Pearson’s correlation (r) for joint kinematics. The values were averaged for all iterations.

		RMSE (°)	Pearson Correlation
		(Mean ± SD)	(Mean ± SD)
**Sagittal**	Elbow flex/ext.	2.03 ± 0.08	0.80 ± 0.02
	Knee flex/ext.	5.94 ± 0.31	0.95 ± 0.01
	Hip flex/ext.	4.52 ± 0.19	0.95 ± 0.00
	Shoulder flex/ext.	6.48 ± 0.40	0.71 ± 0.03
	Trunk tilt	1.94 ± 0.06	0.55 ± 0.03
	Pelvis tilt	1.91 ± 0.06	0.55 ± 0.03
**Frontal**	Hip add/abd.	2.59 ± 0.10	0.70 ± 0.03
	Shoulder add/abd.	3.27 ± 0.25	0.73 ± 0.03
**Transverse**	Trunk rotation	1.72 ± 0.11	0.87 ± 0.025
	Pelvis rotation	2.71 ± 0.09	0.71 ± 0.02

**Table 3 sensors-21-06786-t003:** Results of the validation of the robotic system when compared with a Vicon system. The table shows the accuracy of the system in retrieving the main gait descriptors. The values were averaged for all iterations.

Spatiotemporal and Kinematic Parameters	RMSE (Mean ± SD)	Spatiotemporal and Kinematic Parameters	RMSE (Mean ± SD)
step width (m)	0.021 ± 0.01	pelvis min. tilt (°)	1.67 ± 0.31
left stride length (m)	0.07 ± 0.01	hip max. adduction (°)	1.37 ± 0.35
right stride length (m)	0.05 ± 0.01	hip min. abduction (°)	2.22 ± 0.67
left stride time (s)	0.04 ± 0.01	pelvis max. rotation (°)	2.96 ± 0.97
right stride time (s)	0.06 ± 0.02	pelvis min. rotation (°)	2.02 ± 0.47
right step time (s)	0.03 ± 0.01	hip max. extension during stance (°)	3.60 ± 0.83
left step time (s)	0.026 ± 0.01	hip max. flexion during swing (°)	2.61 ± 0.54
right cadence (steps/min)	4.60 ± 1.27	hip max. flexion during stance (°)	3.70 ± 1.05
left cadence (steps/min)	4.01 ± 1.51	knee initial contact position (°)	4.91 ± 1.33
percentage of foot stance (%)	2.74 ± 0.73	knee position at toe-off (°)	3.29 ± 1.03
percentage of foot swing (%)	1.32 ± 0.29	knee max. flexion in load response (°)	4.52 ± 1.03
trunk max. tilt (°)	1.68 ± 0.45	knee max. flexion during swing (°)	3.39 ± 1.69
trunk min. tilt (°)	2.18 ± 0.35	knee max. extension before heel strike (°)	4.81 ± 1.52
pelvis max. tilt (°)	1.33 ± 0.52		

**Table 4 sensors-21-06786-t004:** Robot’s deviation from the planned trajectory.

	Average Deviation (m)	Standard Deviation (m)	Max. Deviation (m)	Min. Deviation (m)
Experiment 1	0.0446	0.0368	0.1420	0.0018
Experiment 2	0.0427	0.0363	0.1837	0.0021

**Table 5 sensors-21-06786-t005:** Robot’s deviation from the global plan (the planned trajectory) in clinical environments.

	AverageDeviation (m)	Standard Deviation (m)	Max. Deviation (m)	Min. Deviation (m)
Clinical environment A	0.0869	0.0829	0.3193	0.0015
Clinical environment B1	0.0505	0.0476	0.2715	0.0025
Clinical environment B2	0.0389	0.0302	0.1929	0.0004

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
