# Peer review of "ROBOGait: A Mobile Robotic Platform for Human Gait Analysis in Clinical Environments"

_sensors, 2021, doi:10.3390/s21206786_

Round 1

Reviewer 1 Report

This paper presents a very interesting work on the use of mobile robotic platforms for gait monitoring and analysis. The proposal describes the robotic platform ROBOGait detailing its main control and navigation techniques (SLAM, path planning, person following, and lane keeping) together with preliminary results obtained with healthy subjects and MS patients. Results suggest that the proposal is effective at measuring human gait together with representing a markerless and cost-effective solution.

Overall, the paper addresses an interesting topic on rehabilitation robotics and represents a solid contribution to Sensors. The paper is well written, it is easy to read and to follow. Still, my recommendations for the revised versions are:

1. There are no photos of the mobile robot nor the experimental setup. They should be included together with the designs of figures 1, 2, and 10.

2. The paper should include a skeleton image captured by the camera to ease the understanding of the person tracking process.

Minor:

3. Lines 9 and 64: a great experimental stage. Delete "great".

4. For some reason, Figures 5 and 6 seem very low quality.

5. Lines 211 and 239: Be more specific with "presented/detailed later". Rewrite to "presented in Section x.x".

Author Response

Thank you for allowing Major Revisions of our manuscript, with an opportunity to address your comments.

Reviewer 2 Report

 I would like to thank the authors for the interesting manuscript regarding ROBOGait. Mobile robotic platform for human gait
analysis in clinical environments.  Although it is a very interesting topic I do have some concerns regarding the clinical objective and relevance. In general, the rationale of your introduction is not fully addressed by your methodology.  Due to these methodological concerns, it is of course difficult to discuss the impact and clinical relevance of this manuscript which is also missing In the discussion. 
P 2, line 36.37 - I am not convinced that these types of systems are invasive,  after all they can be placed  the markers to the surface of the body, without disturbing the continuity of the tissues of the human body. In addition, they do not have to be placed directly to the patient's body.
line 64-68 - the authors write that the platform has been tested in clinical  environment in a group of patients with MS, but in this paper they ignore the results obtained in this group of patients.
P 5 - Are only the speed and general trajectory adjusted? What in the case of imbalance or pathological gait that may occur in patients after a stroke.
P 9 - Have the authors considered changes in the position of the spine in the event that the person changes shoes, eg women wear shoes with heels?
P 12 - Was the study approved by the local ethics committee?
P 13 - Please explain why in Table 1 does not present data from the ankle and data from other body segments is mainly limited to the sagittal plane, for the remaining planes only selected data are presented?
P 14 - in the case of a curved trajectory, it was limited only to the sagittal plane, I wonder how the body segments behaved in other planes, moreover, the analysis showing the shear forces and the joint torque would be interesting.
Discussion - please refer to your own study results and explain their clinical relevance.
Conclusions - Please explain the impact of this study on the clinical environment.

Author Response

(The authors gave the same response as above.)

Reviewer 3 Report

Review of paper:

ROBOGait. Mobile robotic platform for human gait analysis in clinical environments

Authors:

Diego Guffanti, Alberto Brunete, Miguel Hernando, Javier Rueda, Enrique Navarro

Comments:

The content of the article submitted for review provides an interesting description of the authors' scholarly output, and the aspects addressed in the article fall within the scope of the Sensors Journal theme. Currently, the use of robotic solutions for rehabilitation is becoming a new and still explored engineering challenge in the field of service robotics, hence the topic should be considered timely. Nevertheless, the text of the paper still requires some additional corrections in order to increase the level of content of this study and to make the text more accessible to the Reader.

List of detailed comments bellow:

1. I suggest rebuilding and expanding section 2. State of the art, where the authors more specifically situate some of their research in relation to the work of other authors who are working on converging topics. Citing sources in the text, e.g., [11-19], is insufficient to differentiate how each paper fundamentally differs or converges. Some of the information contained in section 5 Discussion can be used to expand section 2 and precisely define the scientific objective of the paper. Thus, section 5 needs to be revised and rebuilt toward the specific conclusions resulting from the research.

2. Section 3.1 Mechanical design and components - I don't know what is the purpose of including Fig.1 as a CAD visualization of the mobile robot's design, as well as a cross section of Fig. 2 - What for? What does it give to the Reader? In my opinion, this adds nothing to the content of the paper and should rather be replaced with a real photo of the robot system being used in the research environment in interaction with the patient – correct, please.

The paper in my opinion should better describe the laboratory stand and include a description of the limitations of the study.

3. Fig. 6 - Why have the indicated values been taken in the blocks of the diagram? Please comment further in the text.

4. Please address the question of under what time of year the skeleton tracking system surveys were conducted, and whether there were constraints of a research nature in the research environment, such as objects interfering with the correctness of the measurements and building of the quality of the skeleton capture system (e.g. non-uniform lighting, heaters interfering with measurements, etc.).

5. Please address the question to what extent 10 measurements per patient over a period of 15 minutes made the measurements acceptable or not?

6. I propose to combine the results from Fig. 11 and Fig. 12 into one graph so that it is easier to compare the results from Experiment 1 and 2.

Author Response

(The authors gave the same response as above.)

Reviewer 4 Report

The manuscript titled “ROBOGait. Mobile robotic platform for human gait analysis in clinical environments” investigates the reliability of a new markerless gait analysis method based on a robotic controller. This paper provides a detailed explanation of this new robotic approach, even if some points need to be cleared.

This paper is interesting and suitable with the remit and purpose of the journal even if major and minor concerns need to be solved before suggesting for publication.

Abstract

Section words like “Background, Results” can be avoided, especially like results that are repeated. Besides the technical information of the robot, the abstract should provide a general overview of the manuscript. Right now, it does not respect this condition. Authors can clear and improve it.

Keywords

Authors could add “markerless system” and “motion capture”

Introduction

L34-53 This part needs to be reworked because it appears a bit confusing. The classification of different human motion analysis systems should be written more clearly. Authors can use these papers updated to recent approaches:
Technological advancements in the analysis of human motion and posture management through digital devices. https://dx.doi.org/10.5312/wjo.v12.i7.467
Evaluation of the Pose Tracking Performance of the Azure Kinect and Kinect v2 for Gait Analysis in Comparison with a Gold Standard: A Pilot Study. https://doi.org/10.3390/s20185104

L54-68 This section might be better in the Materials and Methods rather than Introduction.

L69 – 76 Section explanation is not required. Conversely, the authors could clarify the aim of this study.

State of art

Authors are citing other studies with the same approach and comparing their robots. However, since this is written initially, readers may find it difficult to understand the difference between robots since the authors’ one has not been explained yet. It is suggested to include this section in the Discussion rather than Introduction part.

Materials and Methods

Figure 1 numbers can be explained as Figure 2.

A section explaining the human motion analysis could be added. The authors explain the robotic features, but low information about motion capture are present. This condition unties the connection to Results because only robotic methods are present rather than human motion. Paragraph “4.1. Validation in a controlled environment” could be moved in M&M section, so results are supported.

Results

Is Table 1 the only one providing data acquired by this robot? This table needs more attention because it should be the aim of this study, right? Results are poor for human motion analysis, notably because authors said this system was compared to Vicon one, but results are absent. The authors have to provide a comparison of results to attest to the reliability of this system. Once readers know that data can be compared with Vicon’s, outside environment and MS analysis can be trusted.

This condition can be complicated, but authors have to decide what results they want to show. If the aim is to provide the validity of this robot, regardless of its field of use, authors may remove human motion data, MS data and show the validity of the robot. If authors intend to show the applicability in clinical settings, human motion data are necessary.

Joint kinematics could be added and compared with Vicon’s ones.

Discussion

L461 Papers or authors need to be appropriately cited, not just the citation number. “The main concern when analyzing the study of who?”. Authors repeated this many times also with different studies, correct it.

L499-500 Microsoft Kinect is no longer produced, but Azure Kinect, yes. So, according to the previous recommended papers, authors should change this statement. Authors can decide to use the camera they please, but for those in the human motion field, this explanation does not justify using the Orbbec Astra camera.

Conclusions

This paragraph does not provide the conclusions of the paper. Secondly, study limitations are not present. Authors have to highlight better the scientific/clinical relevance of the work and include limitations.

References

References’ years are not all bolded. Check it.

Minor checks

Some paragraphs end with a dot. It is not required.

L436 “In The” the T is capitalized.

Author Response

(The authors gave the same response as above.)

Round 2

Reviewer 2 Report

Thank you, I am satisfied

Reviewer 4 Report

The authors followed all the recommended suggestions. The presentation of data compared to Vicon ones improved the topic. My only doubt is about the paragraph "State of art" which is too long. The paper counts 22 pages, a bit excessive.

There are only several editing errors: 

L503 it's bolded

L346-407-474-504-563 authors have to remove the dot at the end of the titles.